# The Development and the Validation of a Novel Dissolution Method of Favipiravir Film-Coated Tablets

**Özge Göktuğ *** , **Ecem Altaş, Gönül Kayar and Mine Gökalp**

Analytical Development Department of R&D Center, Abdi Ibrahim Pharmaceuticals, Istanbul 34538, Turkey; altasecem.3@gmail.com (E.A.); gonul.kayar@abdiibrahim.com.tr (G.K.); mine.uz@abdiibrahim.com.tr (M.G.)
* Correspondence: ozge.goktug@abdiibrahim.com.tr; Tel.: +90-2126226850

**Abstract:** The aim of this study was to develop and validate a dissolution test for favipiravir release in a tablet dosage form using ultra-high performance liquid chromatography (UHPLC). The dissolution method was developed by testing the solubility of favipiravir in media with different pH values. The results demonstrated that the best dissolution was achieved in phosphate buffer with a pH of 6.8. The amount of favipiravir that was released was about 100% after 30 min. The UHPLC method presented linearity (R = 1.000) in the concentration range of 0.044–0.44 mg/mL. The recovery parameter that was achieved ranged from 102.5% to 104.2%. The system suitability, repeatability, and intermediate precision RSD% results were found to be 0.36%, 1.99%, and 2.49%, respectively. In addition to these parameters and results, an F-test was performed using the Minitab 18 Statistical Software program for the intermediate precision and repeatability results. The standard and sample solutions were found to be stable for 2 days in their respective dissolution medium. This analytical method was also found to be selective for favipiravir. In conclusion, a simple and feasible dissolution method with a short run time of 2.5 min was developed and validated successfully. The obtained results demonstrated that the dissolution test developed here is adequate for its purpose and can be applied as the dissolution method for favipiravir in film-coated tablets for release analyses.

**Keywords:** favipiravir; dissolution test; solubility study; method development; validation

## 1. Introduction

COVID-19, also known by its full name, coronavirus 2019, is an infectious respiratory illness that affects many people, which is caused by Severe Acute Respiratory Syndrome Coronavirus 2 (SARS-CoV-2). First discovered in 2019 in Wuhan, China, the disease has spread worldwide since it was discovered, causing the COVID-19 pandemic [1]. A specific and effective drug against COVID-19 has not yet been discovered, and no specific drug has been approved for its treatment [2]. However, various antiviral drugs are being researched to treat COVID-19, and some are being used in clinical trials [3]. Some drugs have been shown to demonstrate a beneficial effect on the virus, such as oseltamivir, ivermectin, lopinavir, ritonavir, remdesivir, favipiravir, ribavirin, and chloroquine and hydroxychloroquine, and have been reported to show potential treatment effects against COVID-19 [4,5]. Specifically, favipiravir, which is an already approved drug to treat influenza in Japan, has been shown to have a positive effect, as have other drugs. According to some studies, it has been shown to have a healing effect that can be observed within a short time despite having severe side effects. Although it is not yet approved as the main anti-viral agent for COVID-19, favipiravir is considered a potential candidate drug [5,6]. In mid-February 2020, a clinical trial using favipiravir as therapy for COVID-19 was initiated and achieved promising results [7]. Recently, the treatment guidelines from many countries and some states in India have included favipiravir in their treatment protocols [8]. Over the past few months, clinical studies have been conducted around the world to evaluate the effectiveness of favipiravir for the management of COVID-19 [9].

As mentioned above, favipiravir is an anti-influenza drug that has shown large spectrum antiviral activity against a variety of other RNA viruses [10]. Favipiravir is an odorless powder that is white to light yellow in color, sparingly soluble in methanol and acetonitrile, and slightly soluble in water and ethanol. Favipiravir has the chemical structure depicted below, and it is chemically described as 6-fluoro-3-hydroxypyrazine-2-carboxamine (Figure 1). The molecular weight of favipiravir is 157.10 g/mol, and its molecular formula is $C_5H_4FN_3O_2$.

**Figure 1.** Chemical structure of favipiravir.

The favipiravir molecule is marketed as under the brand name of Avigan Tablets 200 mg®, which are manufactured by Fujifilm Toyama Chemical Co., Ltd. (Tokyo, Japan) and have been used for the treatment of COVID-19. Dissolution is an important test in formulation development studies and is conducted in order to determine a drug's dosage for properties. Dissolution is an official test that is commonly used as a predictor of in vivo performance that is simulated to evaluate the performance of solid dosage forms, transdermal patches, and suspensions, and it is routinely used in quality control (QC) and research and development (R&D) studies [11]. In the case of generic product development (Favipiravir 200 mg Tablets) in the Abdi Ibrahim Research & Development laboratory, these studies need to be performed for a comparison with the reference product (Avigan Tablets 200 mg®).

There are no studies in the literature providing information about the solubility and BCS information for favipiravir as an active substance. Therefore, the aim of this paper is to present a development and validation study for the dissolution test for film-coated tablets of favipiravir and an ultra-high performance liquid chromatography (UHPLC) method for the quantitation of the drug released from the dissolution test. This method has been validated to demonstrate that the test procedure is suitable for its intended purpose. Method validation will be based on validation parameters such as specificity, linearity and range, precision (system precision and method precision), accuracy, robustness, and solution stability. Validation was performed as per the International Conference on Harmonization (ICH) guidelines [12]. An F-test was performed using the Minitab 18 Statistical Software program for the intermediate precision and repeatability results.

## 2. Materials and Methods

### 2.1. Materials, Reagents and Equipments

Favipiravir reference standard (assigned purity, 99.9%) was supplied from Optrix Laboratories Private Ltd. (Hyderabad, India). Favipiravir Film Coated-Tablets, containing 200 mg of Favipiravir, were developed and produced in Abdi Ibrahim Research & Development laboratory.

HPLC-grade acetonitrile was purchased from Merck (Darmstadt, Germany). Glacial acetic acid, potassium chloride, potassium dihydrogen phosphate, orto-phosphoric acid, sodium acetate anhydrous, boric acid, hydrochloric acid (37%), and sodium hydroxide were Ph. Eur. reagent grade and purchased from Merck (Darmstat, Germany). Purified water was freshly prepared using Elga Purelab equipment (Buckinghamshire, England) for analytical measurements.

Acetate buffer with pH 4.5, phosphate buffer (with pH 3.0, pH 6.0, and pH 6.8), borate buffer with pH 8.0, and 0.1 N HCl were prepared as per the USP and Ph. Eur. [13,14].

Kinetex EVO C18 column with a particle size of 1.7 μm (100 mm × 2.1 mm) was purchased from Phenomenex Inc. (Aschaffenburg, Germany).

Equipment and instruments used in the present study are: Seven Compact S210 model pH meter (Mettler-Toledo, Zurich, Switzerland), AX26DR and XP204 model analytical balance (Mettler-Toledo, Zurich, Switzerland), Transonic 890 model ultrasonic bath (Elma, Singen, Germany), WB14 model shaking water bath (Membert, Schwabach, Germany), Acquity model UHPLC system with binary solvent delivery pump, an autosampler, a photodiode array and ultraviolet detector (Waters Corporation, Milford, MA, USA), Cary 50 model UV spectrophotometer (Varian, Cary, NC, USA) and VK 7010 model dissolution equipment (Varian, Cary, NC, USA). Waters Empower 3 software (Waters Corporation, Milford, MA, USA) was used for data acquisition and processing.

*2.2. Methods*

2.2.1. Filter Compatibility

The function of the filter is the removal of particles from sample solutions that are insoluble and may cause turbidity. The evaluation of the filter must be performed to control whether the active substance is adsorbed by the filter or not [15]. In this study, filter compatibility was evaluated using different filters, i.e., 0.45 μm RC, 0.45 μm PTFE, 0.45 μm PVDF, and 0.45 μm Nylon. Standard solutions were prepared in dissolution media with different pH values, with a final concentration of 0.222 mg/mL, and the solutions were filtered through 0.45 μm RC filter, 0.45 μm PTFE filter, 0.45 μm PVDF filter and 0.45 μm Nylon filter. Unfiltered and filtered standard solutions were injected, and chromatograms were observed.

2.2.2. Solubility Determination

In the early stages of dissolution method development studies, it was important to select the most appropriate dissolution media in order to evaluate the performance of the dosage form. Therefore, solubility data were used as the basis for the selection of the dissolution media for favipiravir.

For favipiravir solubility, the value of the product corresponding to the highest therapeutic dose (1600 mg) taken at one time was studied in 250 mL beakers containing different media [16].

Since the expected solubility values could not be obtained in solubility studies at this concentration (1600 mg/250 mL), solubility studies have been carried out with 200 mg (maximum strength dose) in different media, such as distilled water, acetate buffer with pH 4.5, phosphate buffer with pH 3.0, phosphate buffer with pH 6.0, phosphate buffer with 6.8, borate buffer with pH 8.0, and 0.1 N HCl as per the European Medicines Agency (EMA) [17]. The samples were shaken in a shaking water bath at $37.0 \pm 0.5\ °C$ for 24 h. After 24 h, the samples were taken from the shaking water bath and diluted with relevant dissolution media at a final concentration of 0.222 mg/mL, then filtered and injected into the LC system. The solubility% value of the samples was calculated with a linearity equation, as depicted in Equation (1). Solubility and dissolution% calculations were determined using the following Equations (2) and (3):

$$y = ax + b \tag{1}$$

$$\text{Solubility (mg/mL)} = \frac{A_{Sam} - b}{a} \times S \times 100 \tag{2}$$

$$\text{Dissolution\%} = \frac{A_{Sam} - b}{a} \times \frac{250}{W_{Sam}} \times S \times 100 \tag{3}$$

y       : Area
x       : Concentration (mg/mL)
a       : Slope
b       : Intercept
$A_{Sam}$ : Area of Sample
$W_{Sam}$: Weight of Sample (mg)
S       : Dilution Coefficient

The dose solubility ratio is calculated as follows: highest single therapeutic dose (mg) divided by solubility (mg/mL). Dose (mg)/Solubility (mg/mL) = Dosage Volume (mL). An active substance is considered highly soluble when Dosage Volume is 250 mL or less [18].

### 2.2.3. Selection of Dissolution Volume, Stirring Rate and Apparatus

The theoretical solubility condition value of favipiravir was evaluated with different dissolution volumes. Solubility data obtained from the above calculation were taken as the basis for the selection of a dissolution volume for favipiravir.

The apparatus and stirring rate were determined as per EMA and FDA guidelines [19,20].

### 2.2.4. Dissolution Method Development and Determination of $\lambda_{max}$

After determining the most suitable dissolution medium, the standard solution at 100% concentration (0.222 mg/mL) was scanned in the range of 190–400 nm in 1.0 cm quartz cell, and spectra were recorded to determine the $\lambda_{max}$ of the active substance.

### 2.2.5. Dissolution Test Chromatographic Conditions

The Waters Acquity UHPLC system was utilized for the analysis. Analytical method development and validation were performed on Phenomenex Kinetex EVO C18, $2.1 \times 100$ mm, 1.7 μm stationary phase (Aschaffenburg, Germany). The analysis was carried out in a column with an oven temperature of 35 °C, and the sample temperature was maintained at 25 °C with isocratic conditions using a mobile phase consisting of a mixture of the buffer solution with pH 2.5 and acetonitrile in the ratio 80:20 (v:v). The mobile phase was filtered with 0.2 μm Millipore membrane filter and degassed by sonication. The mobile phase was run at a flow rate of 0.4 mL/min. The injection volume was 1 μL for the blank, standard, and sample solutions. The total chromatographic run time was 2.5 min. Before initiating the analysis, every standard and sample were filtered through 0.22 μm RC filter, and the analysis was monitored at 225 nm.

Preparation of Buffer Solution with pH 2.5: In 1000 mL purified water, 1.36 g of potassium dihydrogen phosphate was dissolved. After dissolving the buffer solution, pH was adjusted to $2.5 \pm 0.05$ with diluted orto-phosphoric acid.

Preparation of Standard Solution ($C_{Favipiravir}$: 0.222 mg/mL): First, 22.2 mg of favipiravir reference standard was weighed into a 100 mL volumetric flask. Afterwards, 5 mL of acetonitrile was added and sonicated for 10 min to dissolve and diluted to volume with dissolution media. It was filtered through 0.22 μm RC filter and transferred to vial.

Preparation of Sample Solution ($C_{Favipiravir}$: 200 mg Favipiravir Tablet/900 mL dissolution media: 0.222 mg/mL): Dissolution testing was performed in compliance with USP 30 [21] using paddles (apparatus II) at 50 rpm, and the bath temperature was maintained at $37.0 \pm 0.5$ °C. Nine hundred milliliters of phosphate buffer with pH 6.8 solution, which needed to be freshly prepared, was used as the dissolution medium. Dissolution samples were collected at 30th minutes due to the highly soluble nature of favipiravir. Samples aliquots were filtered through a 0.22 μm RC filter and analyzed by UHPLC. The cumulative dissolution% was calculated using a standard calibration curve.

### 2.2.6. Dissolution Method Validation

The proposed development method was validated with specificity, linearity and range, system precision, method precision (repeatability and intermediate precision), accuracy, robustness, and solution stability parameters in accordance with the ICH guidelines- [12].

#### Specificity

The specificity test is the ability of the method to measure the analyte response in the presence of other substances or those that are expected to be present. Specificity was examined by analyzing blank (dissolution medium), placebo, standard, and sample solutions. The placebo solution consists of all the excipients (povidone K30, colloidal silicon dioxide, crospovidone CL, sodium starch glycolate type-A vivastar, sodium stearyl

fumarate, and opadry 03A42214 yellow) without the favipiravir active substance. The overlay of blank, placebo, standard, and sample chromatograms were recorded. There could not be any peaks at the retention time of the favipiravir peak from the blank and placebo solutions in chromatograms of standard and sample solutions. The spectrum of the favipiravir peak in the chromatogram obtained from standard and sample solutions could have no interference with other peaks. The purity angle of the favipiravir peak could be less than the purity threshold in chromatograms of standard and sample solutions.

Linearity and Range

The linearity of an analytical method is its ability to elicit test results that are proportional to the concentration of analytes in the samples within a given range. The linearity plot was constructed for favipiravir in the concentration range from 20% to 200% of the standard concentration (0.222 mg/mL). The appropriate dilutions were made from favipiravir stock standard solution prepared in order to achieve 6 different concentrations. The calibration curve was plotted as the concentration of the respective drug solutions versus the peak area at each level. The slope, y-intercept, and correlation coefficient (R) were determined. The correlation coefficient between concentration and areas should not be less than 0.99.

System Precision

System suitability ensures the quality of the method for the accuracy of the results. To determine the system suitability standard solution is prepared at 100% concentration and injected six times. The average, SD, and RSD% for the peak area of favipiravir were calculated. The RSD% of peak areas should be less than 2.0%, theoretical plate count of peaks should be greater than 2000, and symmetry factor of peaks should be within 0.8–1.5.

Method Precision—Repeatability

The repeatability parameter is performed with an optimized dissolution test on six tablets. The precision of the method was demonstrated by calculating RSD% of release% for six measurements. The RSD% of sample results should be less than 5.0%.

Method Precision—Intermediate Precision

The intermediate precision of the method is verified by conducting the precision study using different instruments, different columns of the same make, by different analysts on different days under the same experimental conditions. Six samples from the same batch were prepared and analyzed by the proposed method. The RSD% of sample results between repeatability and intermediate precision parameters should be less than 5.0%.

F-test was performed using the Minitab 18 Statistical Software program for the comparison of the results between repeatability and intermediate precision. The F-distribution was developed by Fisher in order to test the equality of the average values and compare the precision of the measurements. If the *p*-value is less than or equal to the specified significance level alpha, the null hypothesis is rejected; otherwise, the null hypothesis is not rejected. Hence, variances are considered equal [22].

Accuracy

The average recovery% of favipiravir is calculated to represent the accuracy of the method. The accuracy of the proposed method was further assessed by recovery studies at different concentration levels by the standard addition method. The recovery studies were performed at three different levels—10%, 100%, and 120%—of working-level concentration. At each level, three determinations were performed. The recovery% was calculated from the experimental and theoretical amounts. The overall RSD% of samples results should be less than 2.0%.

Robustness

The robustness is the ability of an analytical method to remain unaffected by making small variations in method parameters such as wavelength ($\pm 2$ nm), flow rate ($\pm 0.1$ mL/min), column temperature ($\pm 2$ °C) and stirring rate ($\pm 5$ rpm). The chromatographic parameters like retention time, symmetry factor, theoretical plate count and release% were recorded, and the results were reported.

Solutions Stability

Standard and sample solutions were stored at room temperature (25 °C) and were analyzed by UHPLC for 48 h at specified time intervals to evaluate solution stability in the optimized dissolution medium. The standard and sample solutions are stable until the range similarity of 98.0–102.0% value.

## 3. Results and Discussion

### 3.1. Filter Compatibility

The compatibility of 0.45 μm RC filter, 0.45 μm PTFE filter, 0.45 μm PVDF filter and 0.45 μm Nylon filter was studied. Standard solutions were filtered and analyzed. The similarity% results of peak areas between unfiltered and filtered standard solutions were calculated and tabulated in Table 1. The results were evaluated, and a 0.45 μm RC filter was found suitable for filtration because of the fact that the best performance in all results was obtained from a 0.45 μm RC filter, and it is widely used and less costly [15,23].

**Table 1.** Filter Selection: Area Similarity% Results for Favipiravir Standard Solution.

| Filter <br> Media | Unfiltered | 0.45 μm RC | 0.45 μm PTFE | 0.45 μm PVDF | 0.45 μm Nylon |
|---|---|---|---|---|---|
| 0.1 N HCl | - | 100.6 | 101.2 | 101.1 | 101.6 |
| Phosphate Buffer with pH 3.0 | - | 100.4 | 100.4 | 100.4 | 100.3 |
| Acetate Buffer with pH 4.5 | - | 100.0 | 97.3 | 97.3 | 97.6 |
| Phosphate Buffer with pH 6.0 | - | 97.5 | 97.9 | 98.0 | 98.5 |
| Phosphate Buffer with pH 6.8 | - | 100.4 | 100.7 | 101.4 | 101.8 |
| Distilled Water | - | 100.4 | 100.7 | 101.5 | 102.0 |
| Borate Buffer with pH 8.0 | - | 100.0 | 101.4 | 101.6 | 102.1 |

### 3.2. Solubility Determination

The solubility results of favipiravir in different proposed dissolution media were summarized in Table 2. The data have been demonstrated that favipiravir has low solubility in 0.1 N HCl and phosphate buffer with pH 3.0 media since the dosage volume is higher than 250 mL. Favipiravir active substance is highly soluble in acetate buffer with pH 4.5, phosphate buffer with pH 6.0, phosphate buffer with pH 6.8, distilled water and borate buffer with pH 8.0 media since its dosage volume is less than 250 mL.

**Table 2.** Solubility results of favipiravir in different proposed dissolution media.

| Test Media | Dosage Volume (mL) | Solubility (mg/mL) [1] |
|---|---|---|
| 0.1 N HCl | 380 | 0.526 |
| Phosphate Buffer with pH 3.0 | 290 | 0.689 |
| Phosphate Buffer with pH 6.8 | **245** | 0.815 |
| Acetate Buffer with pH 4.5 | **244** | 0.818 |
| Distilled Water | **243** | 0.822 |
| Borate Buffer with pH 8.0 | **233** | 0.858 |
| Phosphate Buffer with pH 6.0 | **232** | 0.862 |

[1] Average of 3 determinations.

As per the solubility results of favipiravir in different dissolution media in Table 2, distilled water media was not selected because of the fact that the conductivity and pH value are variable. Phosphate buffer with pH 6.0 and borate buffer with pH 8.0 do not reflect the physiological medium recommended in EMA [17]; thus, those media were not selected as the dissolution medium. Acetate buffer with pH 4.5 medium has a close pH value to the pKa value of favipiravir, and the reference product's dissolution results did not reach 100% value for acetate buffer with pH 4.5 medium. The most suitable physiological medium for the solubility study was found as phosphate buffer with pH 6.8. Therefore, phosphate buffer with pH 6.8 was chosen as the dissolution medium.

### 3.3. Selection of Dissolution Volume

According to the solubility results of favipiravir in different dissolution volumes in Figure 2, the sink conditions for favipiravir active substance in 500 mL volume could not be provided when the above-mentioned results of the sink studies in various solutions with different pH values were evaluated. However, the sink conditions were provided for favipiravir active substance in 900 mL volume, acetate buffer with pH 4.5, distilled water, phosphate buffer with pH 6.0, phosphate buffer with pH 6.8, and borate buffer pH 8.0 media. For this reason, 900 mL volume was selected for dissolution test analyses.

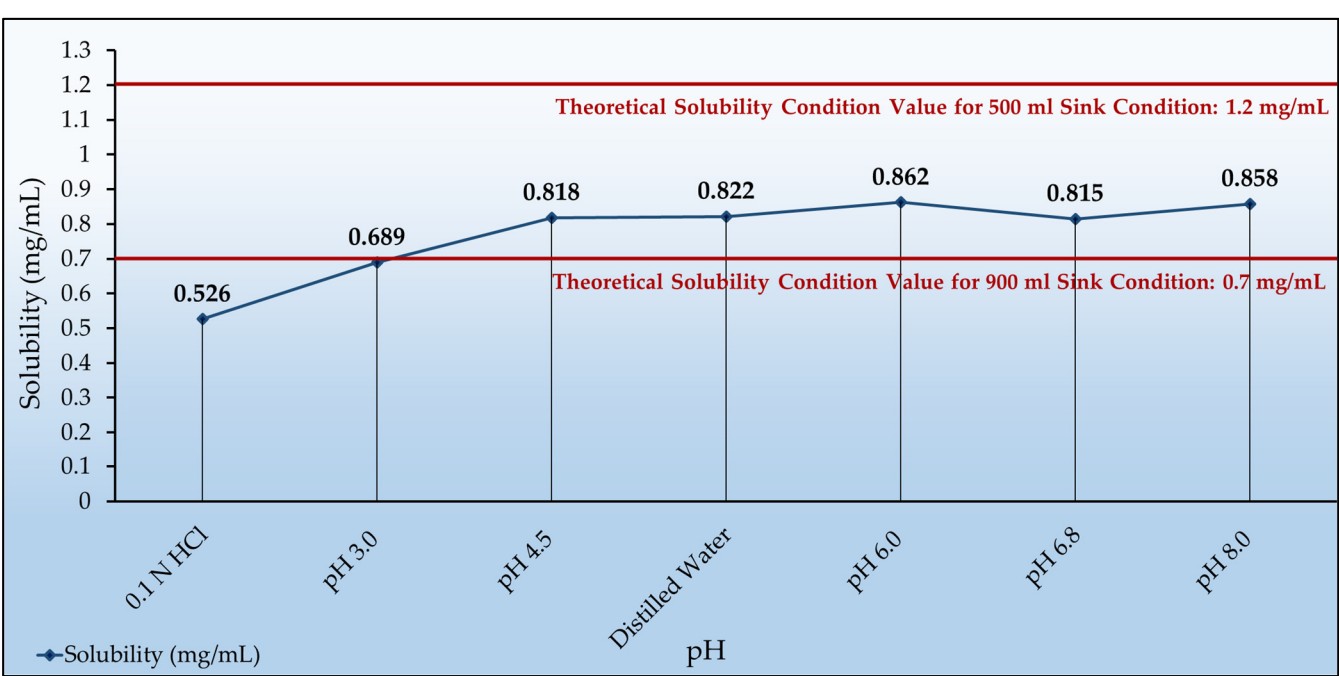

**Figure 2.** Graphical presentation of solubility versus pH of favipiravir in different media and in different dissolution volumes.

### 3.4. Selection of Stirring Rate and Apparatus

EMA and FDA guidelines [19,20] recommended that analytical method development for dissolution studies should be initiated at the speed of 50 rpm with the paddle apparatus due to mild agitation conditions. Therefore, 50 rpm was chosen as the stirring rate for this study.

### 3.5. Dissolution Method Development and Determination of $\lambda_{max}$

The standard solution was scanned in the range of 190–400 nm in a 1.0 cm quartz cell against phosphate buffer with pH 6.8. The UV absorption spectrum of favipiravir shows absorbance peak at 225 nm (Figure 3).

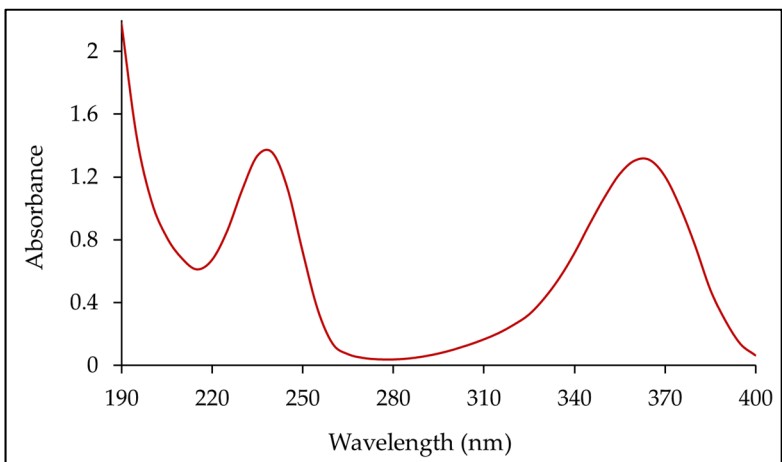

**Figure 3.** UV-Vis spectrum of favipiravir.

The dissolution medium was selected based on the solubility studies. The following dissolution conditions were preferred, and the used analytical method was the UHPLC method. After selecting the best conditions, the validation of the chromatographic method was performed.

*3.6. Dissolution Method Validation*

3.6.1. Selectivity

The specificity of the method was established by injecting blank (dissolution medium), placebo, standard, and sample solutions individually to examine any interference. There are no peaks at the retention time of the favipiravir peak from blank and placebo solutions in chromatograms of standard and sample solutions. The spectrum of the favipiravir peak in chromatograms obtained from standard and sample solutions has no interference with other peaks. The purity angle of the favipiravir peak is found less than the purity threshold in chromatograms of standard and sample solutions. Therefore, the specificity of the method has been proven (Figure 4 and Table 3).

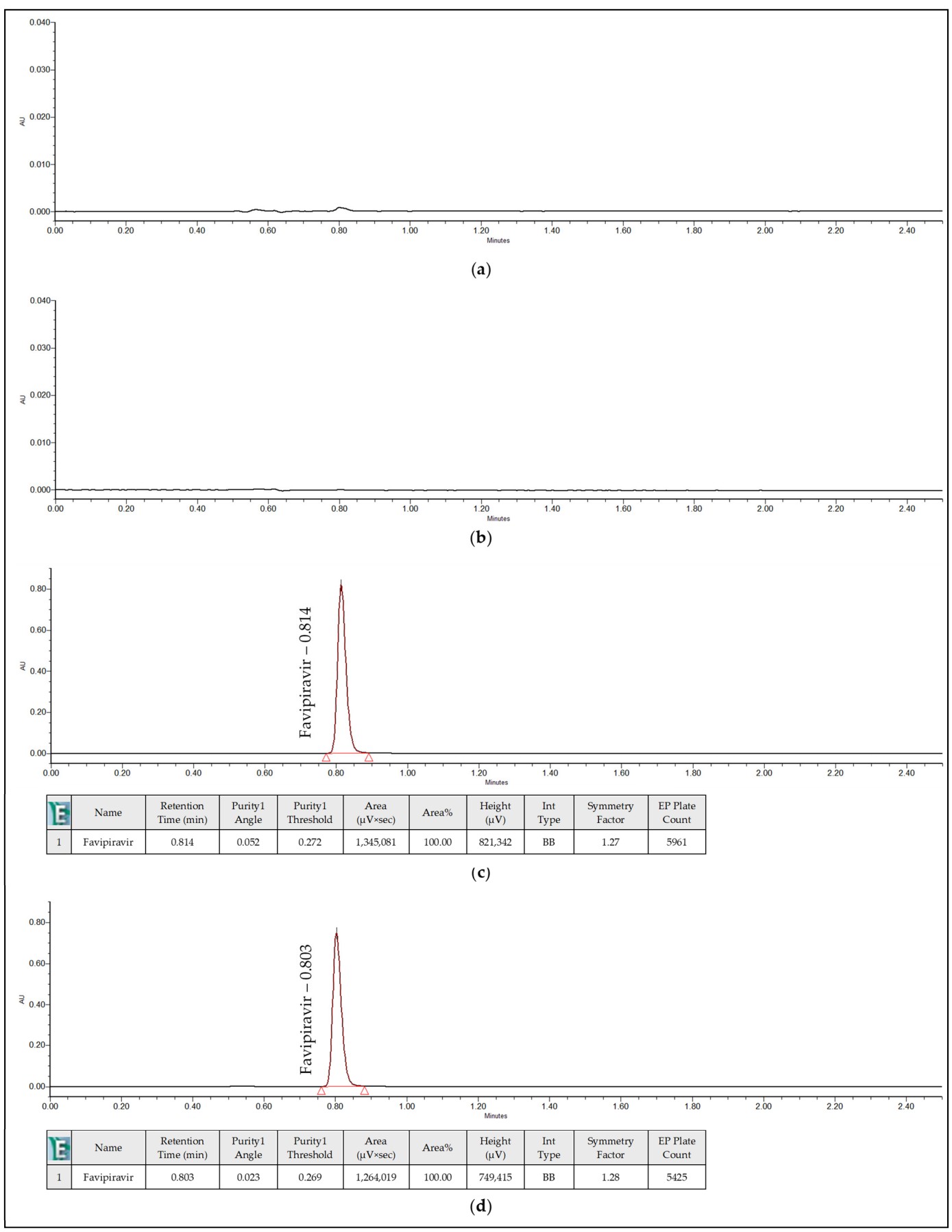

**Figure 4.** Selectivity Chromatogram: (**a**) Blank solution; (**b**) Placebo solution; (**c**) Standard solution; (**d**) Sample solution.

**Table 3.** Results of Selectivity Study.

| Name of Solution | Retention Time (minutes) | Peak Purity | | Purity Criteria |
| --- | --- | --- | --- | --- |
| | | Purity Angle | Purity Threshold | |
| Blank | No peaks | - | - | - |
| Placebo Solution | No peaks | - | - | - |
| Standard Solution | 0.814 | 0.052 | 0.272 | Pass |
| Sample Solution | 0.803 | 0.023 | 0.269 | Pass |

### 3.6.2. Linearity and Range

The range of reliable quantification was set at the concentrations 0.044–0.44 mg/mL. Peak areas and concentrations were subjected to least square regression analysis to calculate the regression equation. Slope, y-intercept, and correlation coefficient (R) values were calculated. For the favipiravir linearity regression equation, y = 6,336,093x + 29,605, and the correlation coefficient was found as R = 1.000 (R > 0.99) (Figure 5 and Table 4). The proposed method has been found as linear.

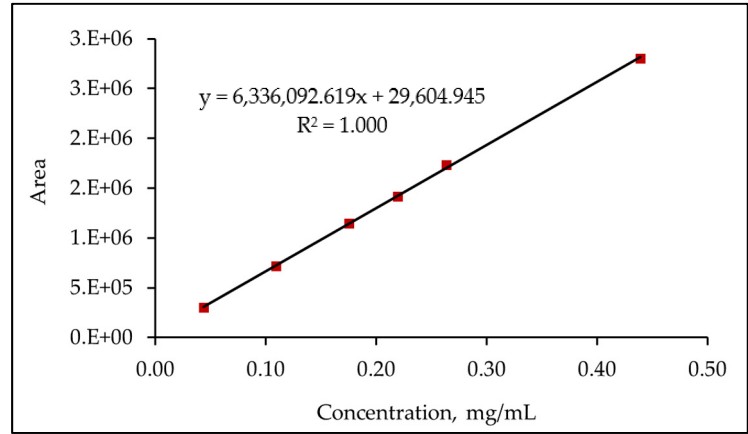

**Figure 5.** Linearity curve of favipiravir.

**Table 4.** Results of Linearity.

| Level% | Concentration (mg/mL) | Area |
| --- | --- | --- |
| 20 | 0.04394610 | 301,721 |
| 50 | 0.10986525 | 718,160 |
| 80 | 0.17578440 | 1,143,999 |
| 100 | 0.21973050 | 1,417,157 |
| 120 | 0.26367660 | 1,733,645 |
| 200 | 0.43946100 | 2,798,675 |
| Corr. Coefficient | 1.000 | |
| Slope | 6,336,092.9159 | |
| y-intercept | 29,604.9454 | |

### 3.6.3. System Precision

The system suitability test was performed by injecting the standard solution containing 0.222 mg/mL in six replicates. The RSD% of the peak area responses and retention times of analytes were determined. Additionally, the symmetry factor and theoretical plate count were calculated. The RSD% of peak areas was found less than 2.0%, theoretical plate count of peaks was above 2000, and symmetry factor of peaks was within 0.8–1.5 (Table 5). The system precision of the method has been proven.

**Table 5.** Results of System Suitability.

| Sample No. | Symmetry Factor | Theorical Plate Count | Retention Time (min) | Area |
|---|---|---|---|---|
| 1 | 1.06 | 6719 | 0.716 | 1,423,910 |
| 2 | 1.07 | 6710 | 0.717 | 1,435,439 |
| 3 | 1.06 | 6693 | 0.716 | 1,420,744 |
| 4 | 1.06 | 6682 | 0.716 | 1,430,521 |
| 5 | 1.06 | 6666 | 0.718 | 1,429,301 |
| 6 | 1.06 | 6701 | 0.717 | 1,429,755 |
| Average | 1.06 | 6695 | 0.717 | 1,428,278 |
| SD | 0.00 | 19.24 | 0.00 | 5203.5 |
| RSD% | 0.38 | 0.29 | 0.11 | 0.36 |

### 3.6.4. Method Precision—Repeatability

The precision of the method was evaluated by analyzing the assay for six individual samples prepared from the same batch as per the proposed method. The average release% and RSD% for the six sample preparations were calculated. The RSD% was found as 1.99% (Table 6). The RSD% (<5.0%) value indicates that the repeatability of the method has been proven.

**Table 6.** Results of repeatability.

| Sample No. | Favipiravir Release% |
|---|---|
| 1 | 103.1 |
| 2 | 103.4 |
| 3 | 99.9 |
| 4 | 98.8 |
| 5 | 102.7 |
| 6 | 99.7 |
| Average | 101.3 |
| SD | 2.02 |
| RSD% | 1.99 |

### 3.6.5. Method Precision—Intermediate Precision

The intermediate precision of the method was verified by conducting the precision study using different UHPLC system and different columns of the same make, by different analysts on different days. Six samples from the same batch were prepared and analyzed by the proposed method. The average release% and the RSD% for the two sets of data were calculated. The RSD% was found as 2.49% (Table 7). The overall RSD% (< 5.0%) value indicates that the intermediate precision of the method has been proven.

**Table 7.** Results of intermediate precision.

| Sample No. | Favipiravir Release% | | |
|---|---|---|---|
| | Analyst-1 Day-1 Instrument-1 Column-1 | | Analyst-2 Day-2 Instrument-2 Column-2 |
| 1 | 103.1 | | 98.6 |
| 2 | 103.4 | | 97.8 |
| 3 | 99.9 | | 95.4 |
| 4 | 98.8 | | 103.3 |
| 5 | 102.7 | | 99.7 |
| 6 | 99.7 | | 100.3 |
| Overall Average | | 100.2 | |
| Overall SD | | 2.49 | |
| Overall RSD% | | 2.49 | |
| F-Test of Significance | | 0.05 | |
| *p*-value | | 0.521 | |

The *p*-value for the F-test was found as 0.521, and the alpha value was selected as 0.05 according to the 95% confidence interval. Since the *p*-value is greater than the alpha value, the results were found as compatible (Table 7).

### 3.6.6. Accuracy

The recovery% of the method was evaluated by performing recovery studies by spiking 10%, 100%, and 120% standard to the placebo solution through the standard addition method. Individual recovery% and overall RSD% were calculated. The recovery% for each level was found between 102.5% and 104.2%, and the RSD% was found as 0.53% (Table 8). The RSD% (<2.0%) value indicates that the accuracy of the method has been proven.

**Table 8.** Results of Accuracy.

| Level% | Sample No. | Recovery% | Average | RSD% |
|---|---|---|---|---|
| 10 | 1 | 103.4 | 103.5 | 0.29 |
| | 2 | 103.7 | | |
| | 3 | 103.3 | | |
| 100 | 1 | 103.9 | 103.6 | 0.67 |
| | 2 | 104.2 | | |
| | 3 | 102.8 | | |
| 120 | 1 | 102.5 | 102.8 | 0.39 |
| | 2 | 102.9 | | |
| | 3 | 103.1 | | |
| Overall Average and RSD% | | | 103.3 | 0.53 |

### 3.6.7. Robustness

The robustness of the method was investigated by changing the instrumental conditions, such as wavelength, flow rate, column temperature, and stirring rate. The results are reported (Table 9). The method was found to be unaffected by small variations, and the robustness of the proposed method has been proven.

**Table 9.** Results of robustness.

| Analysis Name | Favipiravir Release% | Retention Time (min) | Symmetry Factor | Theorical Plate Count |
|---|---|---|---|---|
| Repeatability | 101.3 | 0.717 | 1.06 | 6695 |
| Wavelength: 223 nm | 100.0 | 0.712 | 1.06 | 6535 |
| Wavelength: 227 nm | 100.0 | 0.712 | 1.06 | 6541 |
| Flow Rate: 0.3 mL/min | 99.3 | 0.948 | 1.05 | 7482 |
| Flow Rate: 0.5 mL/min | 99.8 | 0.571 | 1.06 | 5553 |
| Column Temperature: 33 °C | 99.9 | 0.715 | 1.06 | 6592 |
| Column Temperature: 37 °C | 99.5 | 0.711 | 1.06 | 6556 |
| Stirring Rate: 45 rpm | 88.5 | 0.714 | 1.06 | 6485 |
| Stirring Rate: 55 rpm | 99.4 | 0.713 | 1.07 | 6458 |

### 3.6.8. Solutions Stability

In order to evaluate solution stability in the optimized dissolution medium, standard and sample solutions were stored at room temperature (25 °C) and were analyzed by UHPLC for 48 h at specified time intervals. The standard and sample solutions are found to be stable for 48 h as the similarity% is in the range of 98.0–102.0% (Table 10).

**Table 10.** Results of Solution Stability.

| Injection Time (Hours) | Standard Solution | | Sample Solution | |
|---|---|---|---|---|
| | Area | Similarity% | Area | Similarity% |
| Initial | 1,474,726 | - | 1,462,137 | - |
| 24 | 1,466,496 | 99.4 | 1,449,968 | 99.2 |
| 48 | 1,450,231 | 98.3 | 1,433,548 | 98.0 |

## 4. Conclusions

The proposed dissolution method was developed and validated by the UHPLC method for Favipiravir 200 mg Tablets according to the ICH guidelines [12]. The most suitable dissolution medium was selected as phosphate buffer with pH 6.8 based on the solubility studies as QC medium. The suitable conditions of the dissolution test for Favipiravir 200 mg Tablets were obtained using 900 mL of the dissolution medium containing phosphate buffer with pH 6.8 maintained at $37.0 \pm 0.5\,^{\circ}\text{C}$ with paddles (apparatus II) at 50 rpm for 30 min. The method was validated for various parameters, such as specificity, linearity and range, system precision, method precision (repeatability and intermediate precision), accuracy, robustness, and solution stability. All the parameters have met the acceptance criteria. The stability studies were performed, and the sample solutions were found to be stable for 2 days. The validated method was found selective, linear, precise, repeatable, accurate, and robust. Thus, the aforementioned analytical method with a short run time of 2.5 min can be successfully used for routine analysis of samples for Favipiravir 200 mg Tablets.

**Author Contributions:** Conceptualization, Ö.G.; methodology, Ö.G., G.K. and M.G.; software, Ö.G. and E.A.; validation, Ö.G. and E.A.; formal analysis, Ö.G. and E.A.; investigation, G.K. and M.G.; resources, G.K. and M.G.; data curation, Ö.G.; writing—original draft preparation, Ö.G.; writing—review and editing, Ö.G., G.K. and M.G.; visualization, Ö.G. and G.K.; supervision, G.K. and M.G.; project administration, Ö.G., E.A., G.K. and M.G. All authors have read and agreed to the published version of the manuscript.

**Funding:** This research received no external funding.

**Institutional Review Board Statement:** Not applicable.

**Informed Consent Statement:** Not applicable.

**Data Availability Statement:** It will be supplied upon request.

**Conflicts of Interest:** The authors declare no conflict of interest.

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
