# Peer review of "The Development and the Validation of a Novel Dissolution Method of Favipiravir Film-Coated Tablets"

_scipharm, doi:10.3390/scipharm90010003_

Round 1
Reviewer 1 Report
It is important to develop and verify the dissolution method. As the authors mentioned in the Introduction, classifying the BCS is important. However, the authors did not describe the classification of Favipiravir. The authors should describe it.
Line 225-226: What kinds of excipients are involved in the formulation?
Author Response
It is important to develop and verify the dissolution method. As the authors mentioned in the Introduction, classifying the BCS is important. However, the authors did not describe the classification of Favipiravir. The authors should describe it.
Response: Since there is no BSC information about Favipiravir active substance in the literature the solubility study was carried out to select the dissolution medium. If BSC information were available in the literature, it would help us choose the dissolution medium. Therefore, our purpose in this article is not to determine the BSC, but to decide which medium will be used determination of dissolved amount of Favipiravir.
Line 225-226: What kinds of excipients are involved in the formulation?
Response: Excipients have been added.
Reviewer 2 Report
The current paper present the development and the validation of a novel dissolution method for the favipiravir film coated tablets. For the development of the dissolution method, a solubility study of favipiravir in different pH media was conducted. The dissolution medium was selected taking into consideration the determined solubility and the pKa value of favipiravir, but also EMA recommendations. A 900 mL dissolution volume was selected in order to provide sink conditions. The paddle apparatus and a stirring rate of 50 rpm were selected as per EMA and FDA guidelines recommendations. Also a filter compatibility study was conducted. For the quantification of the released amount of favipiravir, a fast isocratic RP-UHPLC method was developed and validated in line with current ICH Q2 (R1) requirements.
The study is well designed and provide an advance in current knowledge as few dissolution methods are available in literature and no compendial analytical methodology or other UHPLC methodologies for the determination of the favipiravir are published. However, the English language can be improved in some parts of the paper. Some minor adjustments are necessary, but also a rewording is required since the sections Results and Discussion are missing.
Specific comments
Title:
As the solubility study is part of the development of the dissolution method, in this case it should be deleted from the title. An option can be: “The development and the validation of a novel dissolution method of favipiravir film coated tablets”.
Abstract:
Line 10: please replace “Chromatographic” with “Chromatography” and “The method” with “The dissolution method”.
Line 13: please replace “The method” with “The UHPLC method”.
Line 20: please replace “an injection time” with “a run time”.
Introduction:
Line 55: please replace “pharmaceutical form” with “dosage form”.
Line 62: please delete the word “study”.
Results: this section is missing and should be completed with data presented in Method section.
Discussion: this section is missing and should be completed with data presented in Method section.
Materials and Methods:
Line 84: please use the official abbreviation for European Pharmacopoeia (Ph. Eur. instead of EP)
Line 89: please present in more detail the liquid chromatographic system.
2.2.5. Dissolution Method Development: As the actual quantification method is a UHPLC method. The UV-Vis Spectrometry is used only for the determination of the wavelength for the UV detection of the chromatographic method. Therefore, the paragraph from line 176 to line 184 should be deleted.
Line 190: some words seem to be missing. With what is the UHPLC system equipped?
Line 241: please replace “verses” with “versus”.
Conclusions:
Line 320: please replace “an injection time” with “a run time”.
Author Response
Title:
As the solubility study is part of the development of the dissolution method, in this case it should be deleted from the title. An option can be: “The development and the validation of a novel dissolution method of favipiravir film coated tablets”.
Response: Title has been corrected.
Abstract:
Line 10: please replace “Chromatographic” with “Chromatography” and “The method” with “The dissolution method”.
Response: It has been corrected.
Line 13: please replace “The method” with “The UHPLC method”.
Response: It has been corrected.
Line 20: please replace “an injection time” with “a run time”.
Response: It has been corrected.
Introduction:
Line 55: please replace “pharmaceutical form” with “dosage form”.
Response: It has been corrected.
Line 62: please delete the word “study”.
Response: It has been corrected.
Results: this section is missing and should be completed with data presented in Method section.
Response: I had shared all my results in the “methods” title in first draft. The format has been changed because I added “results and discussion” title upon your request. You can find recent changes in the word file I uploaded. And, we have corrected the English language according to your request.
Discussion: this section is missing and should be completed with data presented in Method section.
Response: I had shared all my results in the “methods” title in first draft. The format has been changed because I added “results and discussion” title upon your request. You can find recent changes in the word file I uploaded. And, we have corrected the English language according to your request.
Materials and Methods:
Line 84: please use the official abbreviation for European Pharmacopoeia (Ph. Eur. instead of EP)
Response: It has been corrected.
Line 89: please present in more detail the liquid chromatographic system.
Response: It has been corrected.
2.2.5. Dissolution Method Development: As the actual quantification method is a UHPLC method. The UV-Vis Spectrometry is used only for the determination of the wavelength for the UV detection of the chromatographic method. Therefore, the paragraph from line 176 to line 184 should be deleted.
Response: It has been deleted.
Line 190: some words seem to be missing. With what is the UHPLC system equipped?
Response: It has been corrected.
Line 241: please replace “verses” with “versus”.
Response: It has been corrected.
Conclusions:
Line 320: please replace “an injection time” with “a run time”.
Response: It has been corrected.